# Elevated River Inputs of the Total Alkalinity and Dissolved Inorganic Carbon in the Northern Adriatic Sea

**Michele Giani** [1,*] , **Nives Ogrinc** [2] , **Samo Tamše** [2] **and Stefano Cozzi** [3,*]

1 National Institute of Oceanography and Applied Geophysics—OGS, 34010 Trieste, Italy
2 Environmental Sciences Department, Jožef Stefan Institute, 1000 Ljubljana, Slovenia
3 CNR—ISMAR, Marine Sciences Institute, 34149 Trieste, Italy
* Correspondence: mgiani@ogs.it (M.G.); stefano.cozzi@ts.ismar.cnr.it (S.C.)

**Abstract:** The response of coastal systems to global acidification depends strongly on river inputs, which can alter the total alkalinity ($A_T$) and dissolved inorganic carbon (DIC) in seawater. The northern Adriatic Sea (NAd) is a shallow continental shelf region that currently receives about 15% of the total freshwater input in the Mediterranean Sea, where the role of riverine discharges on the carbonate system has been poorly studied. In particular, river discharges can alter the carbonate system in the sea, affecting both the equilibrium chemistry and biological processes. For the main rivers flowing into the NAd (the Po, Adige, Brenta, Piave, Livenza, Tagliamento, Isonzo, Timavo and Rižana), data were collected for the pH, concentrations of the total alkalinity ($A_T$), $Ca^{2+}$ and $Mg^{2+}$ and the isotopic ratio of stable carbon in the dissolved inorganic carbon ($\delta^{13}C_{DIC}$). The DIC fluxes were estimated using the THINCARB (THermodynamic modeling of INOrganic CARBon) model for the compilation of the $A_T$ and pH data. The results show that the total transport of the $A_T$ in the rivers was 205 Gmol yr$^{-1}$ while the transport of the DIC was 213 Gmol yr$^{-1}$, of which about 70% was from the Po River. About 97% of the DIC in the river waters was in the form of bicarbonates. The high $Mg^{2+}/Ca^{2+}$ ratios indicate that dolomite weathering is predominant in the Adige, Piave, and Livenza river basins, while lower ratios in the Timavo and Rižana rivers indicate a greater proportion of calcite. The mean $\delta^{13}C$-DIC value was estimated to be $-10.0 \pm 1.7$ ‰, a value nowadays considered typical for the DIC flux inputs in oceanic carbon cycle modeling. The DIC flux depends on the mineral weathering and biological activity in each river basin. However, these natural processes can be modified by anthropogenic disturbances that should be better quantified.

**Keywords:** alkalinity; dissolved inorganic carbon; carbonate system; stable carbon isotope; rivers; runoff; time series; Adriatic Sea; Mediterranean Sea

## 1. Introduction

Carbon is continuously displaced along the aquatic land–ocean continuum, which includes freshwaters, estuaries and coastal areas [1,2]. It is transferred in both inorganic (bicarbonates and dissolved $CO_2$) and organic (dissolved and particulate) forms along these environments. Of the 0.9 Pg C transported annually by rivers to the world ocean, about 0.5 Pg C is organic and 0.4 Pg C is inorganic [3].

In freshwater, dissolved inorganic carbon (DIC) is present in three species: carbonic acid, including dissolved $CO_2$ ($H_2CO_3$), bicarbonate ($HCO_3^-$) and carbonate ($CO_3^{2-}$) ions. The total alkalinity is defined as $A_T$ = [$HCO_3^-$] + 2[$CO_3^{2-}$] + the other weak bases that can be converted to acidic forms at the $A_T$ equivalence point (pH < 4.5) through HCl titration [4]. Due to the pH levels in freshwater systems, DIC mainly occurs in the form of bicarbonate ions ($HCO_3^-$) [5].

$CO_2$ in the atmosphere and in the soils (i.e., produced by the bacterial decomposition of organic matter) can be converted into organic carbon through biochemical synthesis or removed through chemical rock weathering. This $CO_2$ is then carried by rivers to the

ocean as particulate organic carbon (POC), dissolved organic carbon (DOC) and DIC. At the same time, continental erosion and rock dissolution cause a natural transfer of carbonates contained in rock from the land to the ocean [6,7].

The chemical processes that occur during the weathering of rocks depend on the rock's composition. The hydrolysis of silicate rocks to form $HCO_3^-$ requires the absorption of $CO_2$ from an external source (i.e., atmospheric or soil $CO_2$), whereas the dissolution of carbonate rocks (mainly calcite, $CaCO_3$; and dolomite, $CaMg(CO_3)_2$) produces a double amount of $HCO_3^-$, half from the rock and half from external sources [8]. In the world's rivers, 64% of the total $HCO_3^-$ comes from $CO_2$ in the soil and 34% from the weathering of carbonate and silicate rocks [6].

On millennial scales, the uptake of atmospheric $CO_2$ is balanced by carbonate sedimentation in the oceans and by its subsequent return to the atmosphere due to tectonic activity and volcanism [9,10]. However, the natural transport of carbon from the land to the oceans is increasingly affected by human disturbances acting on freshwater ecosystems. These include acid rainfalls enhanced by anthropogenic $CO_2$ emissions in the atmosphere, air pollution, changes in land use in watersheds and alterations in land surface hydrology [11,12]. For example, it has been shown that the urbanization of river basins can increase the DIC and DOC concentrations in freshwater as a result of water quality degradation, leading to $CO_2$ partial pressures ($pCO_2$) in freshwater systems that often exceed the atmospheric equilibrium [13]. These perturbations cause uncertain impacts on the coastal areas and feedbacks on the climate [2,7,14].

In the Mediterranean Sea, the $A_T$ is significantly higher than in the other oceans, and it is linearly correlated with salinity [15]. This correlation is positive in the open sea regions of the Mediterranean where evaporation is higher than precipitation. Negative correlations can be found in marginal areas, where river water strongly contributes to the increase in the $A_T$ [16–20]. The northern Adriatic Sea (NAd) is a marginal sea that receives a relevant contribution of the $A_T$ from rivers [18], which increases its concentration in coastal waters [21,22]. The $A_T$ and DIC of the NAd ($A_T$ mean 2657–2685 $\mu$mol $kg^{-1}$; DIC mean: 2350–2378 $\mu$mol $kg^{-1}$) [22] are 15–17% higher than the global ocean values ($A_T$ mean: 2302.7 $\mu$mol $kg^{-1}$ and DIC mean: 2033.7 $\mu$mol $kg^{-1}$) [23]. Therefore, to better assess the fluxes of $CO_2$ between the atmosphere and sea in northern Adriatic continental shelf, it is essential to quantify the riverine $A_T$ and DIC discharges.

The impact of four rivers (the Isonzo, Timavo, Rižana and Dragonja) on the Gulf of Trieste was investigated by Tamše et al. [24], who estimated a riverine discharge of $1.03 \times 10^{11}$ g DIC $yr^{-1}$ (8.58 Gmol $yr^{-1}$) in 2007, with the Isonzo River being the main contributor. Based on the $\delta^{13}C_{DIC}$, the authors estimated that the contribution of the rivers to the DIC content in the Gulf of Trieste reached up to 16% during spring. They also pointed out that these rivers had a higher carbonate weathering intensity than the global average due to their calcareous catchments.

Cioce et al. [25] and Fossato [26] found that the $A_T$ was positively correlated with the $Ca^{2+}$ and $Mg^{2+}$ concentrations in the Po River and inversely correlated with the freshwater discharge. Analogous correlations were found for the Adige River [27]. They estimated that the $HCO_3^-$ contributes approx. 90–95% of the $A_T$.

Despite these publications, river discharges of the $A_T$, DIC and $HCO_3^-$ in the NAd has been poorly studied, and even recent estimates [18] were still based on historical data that do not reflect the current environmental conditions in this region. In particular, a balance has never been established for the northern shelf area, which is the most affected by riverine inputs. The aim of this study is to (i) create a compilation of river discharge data for the carbonate system parameters in NAd for recent years, (ii) analyze the data in terms of the main processes currently driving these parameters and (iii) estimate the DIC and $A_T$ discharges in the NAd region.

This information is important for the management of valuable river–ocean resources, as well as a better understanding of the underlying processes at local and regional scales for global ocean acidification research.

## 2. Study Area

The northern Adriatic Sea (NAd), delineated in Figure 1, is a shallow continental shelf with a surface of about 9930 km$^2$, a volume of 266 km$^3$ and a mean depth of 27 m [28]. This sea area collects runoff from a catchment area of about 110,600 km$^2$, of which 67% of the extent is contributed by the Po River basin [29,30]. The largest freshwater runoff occurs along the west coast, through the Po, Adige and Brenta rivers (84%), while the northern and eastern rivers are less important (10% and 6%, respectively). The NAd drainage basin is a highly industrialized region shared by Italy, Slovenia, Switzerland and Croatia, with large urban settlements, intensive agriculture and a population of about 20 million of inhabitants [29,31,32].

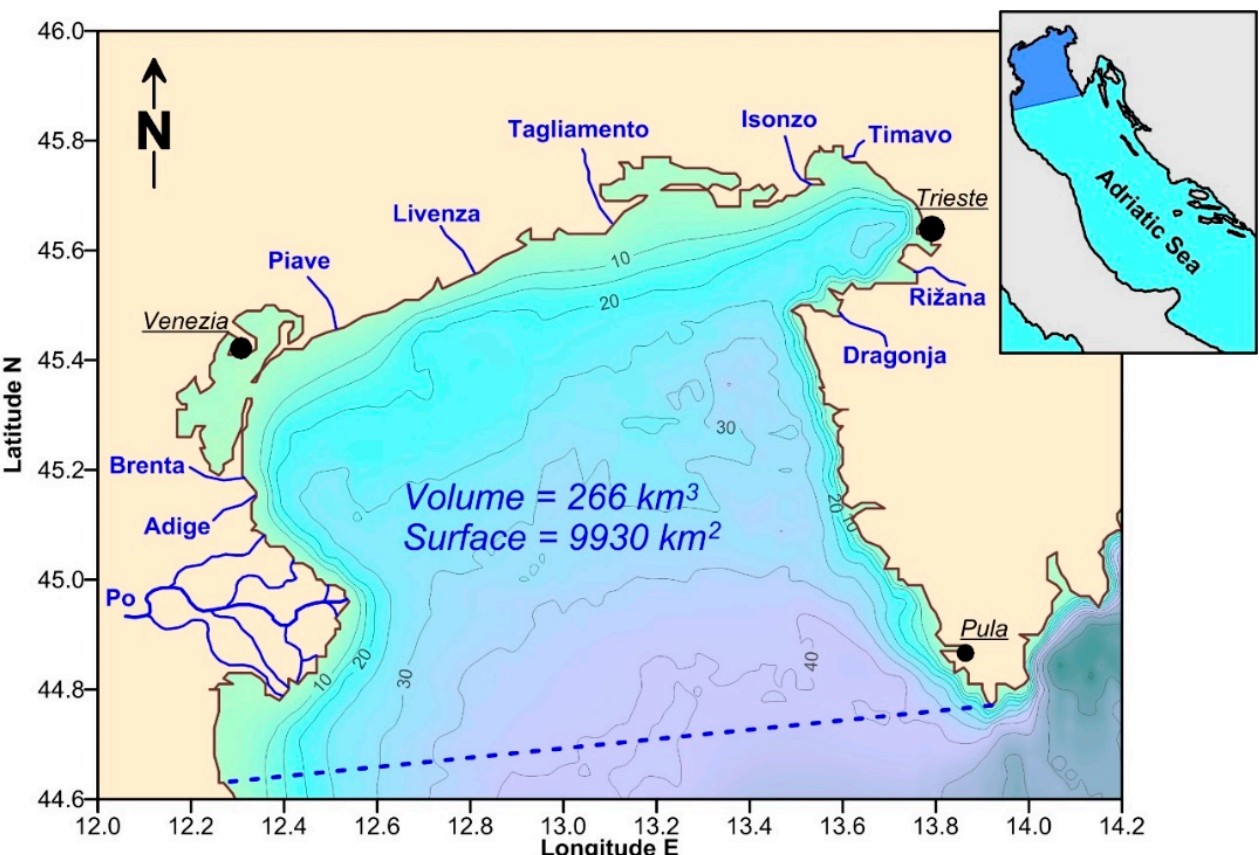

**Figure 1.** Northern Adriatic Sea (NAd) and the rivers (names in blue fonts) considered in the present study.

In the NAd, the annual cycle of runoff is essentially characterized by two periods of high runoff in the spring and fall, due to seasonal precipitation peaks and snowmelt in the mountainous regions. However, strong interannual and decadal fluctuations of discharge have been observed, with low discharges prevailing in most of the early 2000s and a highly fluctuating discharge during the 2010s [28,33]. These long-term fluctuations in the discharges of the NAd watershed are mainly influenced by the effects of climate change on the precipitation, evapotranspiration and melting of snow and glaciers [34].

River discharges carry a large variety of pollutants, such as heavy metals, organic and inorganic compounds and nutrients, which are dispersed in coastal marine environments. The large discharges of total nitrogen (median of 181 ktons yr$^{-1}$) and total phosphorus (10 ktons yr$^{-1}$) are mostly anthropogenic in origin. Nitrogen emissions are primarily due to diffused sources, such as livestock and agricultural activities, while phosphorus emissions are primarily from point sources, such as municipal and industrial wastewater discharges [29,31]. Nutrient loading from rivers have greatly affected the NAd

marine ecosystem in the past, contributing to the occurrence of eutrophic and dystrophic events [32,35].

## 3. Materials and Methods

### 3.1. Freshwater Discharges

The freshwater discharges of the NAd rivers were calculated using the daily averaged flow series ($m^3 \ s^{-1}$) provided, as part of their institutional activity, by the Environmental Protection Agencies (ARPA) of the Regions Emilia Romagna, Veneto and Friuli Venezia Giulia (URL: http://www.arpa.emr.it/; http://www.arpa.veneto.it/; http://www.arpa.fvg.it/, accessed on 15 February 2021) and by a water management company (AcegasApsAmga S.p.A., Trieste, Italy; URL: http://www.acegasapsamga.it/, accessed on 22 February 2021) for the Timavo River. The daily averaged flows were calculated through the continuous recording of the hydrometric heights in the gauge stations, which were then transformed into the flow using the discharge equations calculated in each river section of interest. These data were used to identify the river regime in concomitance with the sampling of the chemical parameters (Table A1) and to calculate the annual integrated river water discharges (Tables A2 and A3). The gauge stations were located at the closure of each river drainage basin for the Po (Pontelagoscuro station), Adige (Boara Pisani station), Piave (Ponte di Piave station), Livenza (Meduna di Livenza station), Isonzo (Slovene: Soča; Pieris station) and Timavo (Bocche di Timavo station). The water flow at the Brenta River mouth was calculated as the sum of the main stem flow (Barziza station) and its tributaries: the Bacchiglione River (Montegalda station) and the Gorzone Channel (Stanghella station). The long-term series of flow rates were not available at the river mouth for the Tagliamento River. Instead, the discharge was evaluated in the station of Pioverno, located at the end of its mountainous drainage basin [28]. The water flows of the Rižana (Kubed station) and Dragonja (Podkaštel station) rivers were published by the Environmental Agency of the Republic of Slovenia (ARSO; URL: http://www.arso.gov.si/en/, accessed on 8 February 2021).

### 3.2. Physical and Chemical Parameters

The physical and chemical parameters were collected at the same gauging station locations or at points closer to the river mouths (Table S1) to provide the best estimate of the concentrations of the chemical components discharged into the sea. This dataset was derived from a research activity by the Jožef Stefan Institute that monitored the activities of a water management company (AcegasApsAmga S.p.A., Trieste, Italy) and from the ARPA of the Regions Emilia Romagna, Veneto and Friuli Venezia Giulia. The chemical sampling covered all the seasons, with a 23% distribution of the data in winter, 28% in spring, 24% in summer and 26% in autumn.

The temperature, conductivity and pH were measured in the field using portable thermal conductivity meters and pH meters. The pH data provided by the Jožef Stefan Institute were measured using a Corning 315 high-sensitivity portable pH meter equipped with an Orion Ross combination glass electrode and a thermometer. The pH was calibrated using buffer solutions that were traceable to NIST standards with the pH values of 4.01 and 7.01 (Hanna Instruments, Inc., Woonsocket, RI, USA), obtaining a precision of ±0.02 pH units. The pH data collected during the monitoring activities (ARPA and AcegasApsAmga S.p.A., Italy) were measured according to a comparable reference method [36] using portable pH meters equipped with combination glass electrodes, temperature correction and two calibration buffers, achieving a typical precision of ±0.02 pH units. All the pH measurements were given on the NBS scale.

The $A_T$ of the river water samples was determined through potentiometric titration, using a pH glass electrode and a titrating solution of HCl 0.05 N standardized against $Na_2CO_3$ primary standard solutions with a precision of ±1% [24,36]. The concentrations ($\mu mol \ L^{-1}$) of calcium ($Ca^{2+}$) and magnesium ($Mg^{2+}$) ions in the river water were deter-

mined using Flame Atomic Absorption Spectrometry or inductively coupled plasma optical emission spectroscopy (ICP-OES) [24,36], with an estimated precision of ±2%.

The estimates of the DIC and $HCO_3^-$ concentrations in the river water were calculated applying the THINCARB software [37] to the compilation of the $A_T$ and pH data of the main rivers flowing into the NAd (Table S1) during 2010–2019. A total of 880 data points were collected for the $A_T$ and the carbonate system in the rivers.

The samples for the determination of the stable carbon isotope ratio in the dissolved inorganic carbon ($\delta^{13}C_{DIC}$) were collected in the main streams of the rivers in evacuated 12-mL Exetainer® glass tubes (Labco Limited, Ceredigion, UK) to prevent gas exchange with the atmosphere. The $\Delta^{13}C_{DIC}$ was determined by injecting aliquots of the sample into the evacuated septum tubes containing phosphoric acid. The released $CO_2(g)$ was then analyzed using a continuous-flow IRMS Europa 20–20 (Crowe, UK) with an ANCA-TG module for trace gas separation. To determine the optimal extraction procedure for the water samples, two $Na_2CO_3$ standard solutions were prepared with a known $\delta^{13}C_{DIC}$ value of $-10.8 \pm 0.2$‰ and $-4.1 \pm 0.2$‰, respectively.

*3.3. Data Processing*

The annual transport of the DIC, $HCO_3^-$ and $A_T$ (F; mol yr$^{-1}$) was estimated only for the major rivers for which a complete data series was available (i.e., the Po, Adige, Brenta, Piave and Livenza rivers). The transport was calculated using the equation based on the discharge weighted means of the daily loads

$$F = [ \sum_{i=1}^{n}(C_i \cdot Q_i) \ / \ \sum_{i=1}^{n} Q_i] \cdot Q_{yr} \qquad (1)$$

where $C_i$ and $Q_i$ are the concentration of the parameter (mol m$^{-3}$) and the flow (m$^3$ s$^{-1}$) for each day of sampling "$i$" during the year ($n \geq 4$) and $Q_{yr}$ is the annual water discharge (m$^3$ yr$^{-1}$). This equation allows for the best weighting of the data to compensate for biases due to the flow variability in the concomitance of the sampling and the different time resolutions of the chemical and flow data series [28]. For the other chemical parameters, such as the $Mg^{2+}/Ca^{2+}$ ratios and the $\delta^{13}C_{DIC}$, all the available data were considered, including those referring to the minor rivers.

The average isotopic signature of all NAd rivers ($\delta^{13}C_{DIC,R}$) was estimated through weighing the isotopic signature of each river "k" ($\delta^{13}C_{DIC,k}$) by its annual average DIC flux ($F_{DIC,k}$), as shown in Equation (2) [38].

$$\delta^{13}C_{DIC,R} = \frac{\sum_{k=1}^{n} F_{DIC,k} \cdot \delta^{13}C_{DIC,k}}{\sum_{k=1}^{n} F_{DIC,k}} \qquad (2)$$

The importance of the annual discharges of the freshwater $A_T$, DIC and $HCO_3^-$ by the NAd rivers was derived by comparing these fluxes with the relevant budgets in the NAd basin and calculating their turnover rates [28]. For freshwater, the turnover rate was calculated as the ratio between the annual freshwater discharge and the volume of the NAd sea area, as shown in Figure 1. For the carbonate system parameters, the inputs from the rivers were compared with the budgets of these components in the sea, which were calculated using the reference mean values in seawater: salinity = 37.32 ± 1.32, $A_T$ = 2745.4 ± 34.5 μmol L$^{-1}$, DIC = 2428.4 ± 43.5 μmol L$^{-1}$ C, $HCO_3^-$ = 2181.9 ± 73.6 μmol L$^{-1}$, pH in situ = 8.114 ± 0.077, $[A_T - DIC]$ = 321.6 ± 131.9 μmol L$^{-1}$. These average values were determined from a dataset (Acid.it Project; n. data = 377) collected during 16 cruises carried out at bimonthly intervals, from December 2014 to January 2017, along one transect from the east to the west that was representative of both high-salinity offshore waters and fresher waters within the NAd coastal fronts [22].

## 4. Results and Discussion

### 4.1. Chemical Characteristics of the NAd River Waters

The rivers of the NAd showed high $A_T$ concentrations, with mean values of $3115 \pm 536$ µmol L$^{-1}$ for the Po River, the main freshwater source in this continental shelf, to extremely high values in some smaller karst rivers such as the Timavo, Rižana and Dragonja (Table 1 and Table S1). Based on the pH levels (7.50–8.17), the $HCO_3^-$ was the predominant form of dissolved inorganic carbon in the river water (2613–5139 µmol L$^{-1}$). The $Ca^{2+}$ concentration was in the range of 860–2131 µmol L$^{-1}$, while the $Mg^{2+}$ concentration was in the range of 306–638 µmol L$^{-1}$, except for the Dragonja River, which had very high concentrations of these two elements at the mouth of the river due to saltwater intrusion. The mean values of the $[A_T - DIC]$ in the river waters were negative for the Po, Brenta, Piave, Timavo and Dragonja, indicating a frequent excess of the DIC with respect to the $A_T$, which originated from the $CO_2$ sovrasaturation in freshwater systems. In the other rivers, the $[A_T - DIC]$ was slightly positive.

**Table 1.** Descriptive statistics of the parameters measured in the river waters (data in Table S1, since the 2000s): number of data (N), mean (AV) and standard deviation (SD).

| River | | pH | $A_T$ | DIC | $HCO_3^-$ | $[A_T - DIC]$ | $Ca^{2+}$ | $Mg^{2+}$ | $\delta\,^{13}C_{DIC}$ |
|---|---|---|---|---|---|---|---|---|---|
| | | | µmol L$^{-1}$ | µmol L$^{-1}$ | µmol L$^{-1}$ | µmol L$^{-1}$ | µmol L$^{-1}$ | µmol L$^{-1}$ | (‰) |
| Po | N | 122 | 122 | 122 | 122 | 122 | 121 | 82 | 3 |
| | AV | 7.78 | 3115 | 3205 | 3023 | −89 | 1456 | 480 | −9.63 |
| | SD | 0.32 | 536 | 534 | 531 | 164 | 229 | 88 | 0.35 |
| Adige | N | 239 | 237 | 237 | 237 | 237 | 25 | 25 | 15 |
| | AV | 8.03 | 2305 | 2260 | 2190 | 46 | 1056 | 489 | −6.95 |
| | SD | 0.21 | 901 | 887 | 865 | 58 | 357 | 361 | 0.70 |
| Brenta | N | 123 | 123 | 123 | 123 | 123 | 15 | 15 | 2 |
| | AV | 7.98 | 4473 | 4578 | 4428 | −105 | 1346 | 519 | −8.95 |
| | SD | 0.18 | 1823 | 1855 | 1804 | 70 | 272 | 160 | 1.20 |
| Piave | n | 122 | 122 | 122 | 122 | 122 | 15 | 15 | 6 |
| | AV | 8.05 | 4271 | 4352 | 4224 | −80 | 1632 | 850 | −7.82 |
| | SD | 0.17 | 1549 | 1561 | 1522 | 78 | 182 | 486 | 1.73 |
| Livenza | n | 88 | 88 | 88 | 88 | 88 | 7 | 7 | - |
| | AV | 8.02 | 4804 | 4766 | 4614 | 38 | 1525 | 683 | - |
| | SD | 0.15 | 1643 | 1670 | 1606 | 131 | 270 | 112 | - |
| Tagliamento | N | 47 | 7 | 7 | 7 | 7 | 44 | 44 | 10 |
| | AV | 8.05 | 3461 | 3454 | 3369 | 8 | 1872 | 879 | −7.13 |
| | SD | 0.16 | 190 | 193 | 185 | 19 | 207 | 109 | 0.48 |
| Isonzo | N | 90 | 77 | 76 | 76 | 76 | 18 | 18 | 5 |
| | AV | 7.99 | 3695 | 3658 | 3537 | 84 | 1233 | 375 | −8.12 |
| | SD | 0.18 | 548 | 561 | 533 | 419 | 236 | 119 | 1.20 |
| Timavo | N | 196 | 63 | 63 | 63 | 63 | 149 | 149 | 12 |
| | AV | 7.49 | 4016 | 4408 | 3950 | −393 | 1843 | 281 | −11.69 |
| | SD | 0.20 | 276 | 424 | 277 | 249 | 172 | 60 | 0.61 |
| Dragonja | N | 5 | 5 | 5 | 5 | 5 | 4 | 4 | 4 |
| | AV | 7.50 | 5247 | 5875 | 5144 | −628 | 2405 | 2578 | −11.60 |
| | SD | 0.51 | 959 | 766 | 939 | 761 | 447 | 1724 | 0.27 |
| Rižana | N | 5 | 5 | 5 | 5 | 5 | 5 | 5 | 4 |
| | AV | 8.19 | 4429 | 4410 | 4309 | 19 | 2044 | 244 | −11.33 |
| | SD | 0.09 | 325 | 335 | 326 | 13 | 65 | 25 | 0.22 |

The stable isotopic composition of the dissolved inorganic carbon showed significantly lower values for the Dragonja, Rižana and Timavo than for the other major NAd rivers. The values of the $Ca^{2+}$, $Mg^{2+}$ and $HCO_3^-$ were high compared to the global weighted average

concentrations (370, 140 and 870 μmol L$^{-1}$, respectively) [6]. The Ca$^{2+}$ and Mg$^{2+}$ in the river waters were primarily derived from rock weathering. The Ca$^{2+}$ sources consisted of carbonate rocks, such as calcite and dolomite, with a minor contribution from Ca–silicate minerals. Dolomite is also a source of dissolved Mg$^{2+}$ in the same ratio as Ca$^{2+}$. Calcite and dolomite occur almost exclusively in sedimentary rocks and contribute, on average, 65% of the Ca$^{2+}$ concentration in river water [6].

### 4.2. Mg$^{2+}$/Ca$^{2+}$ Molar Ratios in River Waters

The mean Mg$^{2+}$/Ca$^{2+}$ molar ratios ranged from 0.12 to 1.07, indicating a relevant variation in the contribution of calcite and dolomite dissolution in the different watersheds (Figure 2, Table A4). In particular, they showed a greater contribution of calcite-to-rock weathering for the Po and Isonzo rivers compared to the other major NAd rivers, and a very high contribution for the karst rivers such as the Timavo and Rižana. The catchment area of the Po and its tributaries covers a large part of northern Italy and is surrounded by the Alps in the north and the Apennines in the south. The Apennines are the most affected by rock weathering and sediment transport, where many easily erodible marine sedimentary rocks rich in calcite are found [39,40]. On the other hand, the Isonzo catchment develops in the Eastern Alps, where calcite and dolomite rocks are predominant, and in a karst region, where Cretacean carbonates predominate, before flowing through the plain [41]. The Timavo and Dragonja catchments are smaller and completely embedded in a karst area (Figure 2).

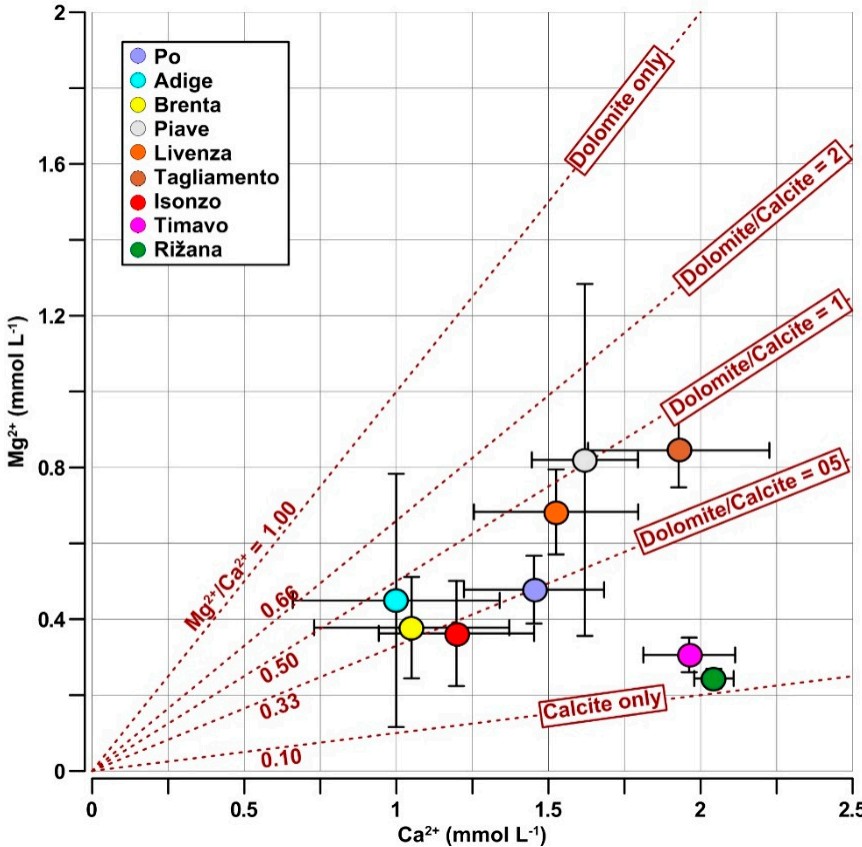

**Figure 2.** Mg$^{2+}$/Ca$^{2+}$ molar ratios in river water (mean values and standard deviation) compared to theoretical values expected from weathering of carbonate rocks (calcite and dolomite). The data of Dragonja River are not included in the figure because the Mg$^{+2}$/Ca$^{+2}$ ratio (=1.07) was strongly influenced by the intrusion of salt water at the river mouth.

It should be noted, however, that the relationships between Mg$^{2+}$ and Ca$^{2+}$ and HCO$_3$$^-$ discharges can be further decoupled by other environmental factors. An overflow

of bicarbonates may result from enhanced rock weathering due to acid precipitation, primarily due to the deposition of sulfuric and nitric acids of anthropogenic origin, while outgassing and primary production reduce the $HCO_3^-$ concentrations in river waters relative to the $Mg^{2+}$ and $Ca^{2+}$ concentrations [6]. The relationships between $Ca^{2+}$ and $Mg^{2+}$ relative to $HCO_3^-$ revealed important differences among the NAd rivers. For the major rivers, from the Po to the Isonzo, the $Mg^{2+}$ concentration was linearly related to the $HCO_3^-$ concentration with a ratio of 0.18, indicating a non-negligible contribution of dolomite weathering to the $HCO_3^-$ transport (Figure 3a, Table A1).

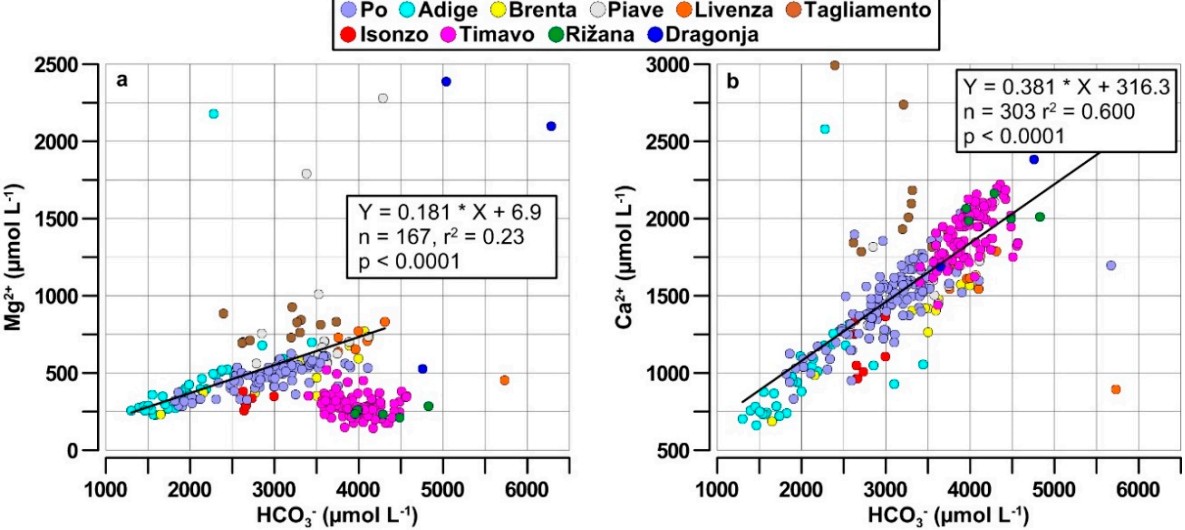

**Figure 3.** Relationships of the concentrations of (**a**) $Mg^{2+}$ and (**b**) $Ca^{2+}$ versus $HCO_3^-$ (µmol L$^{-1}$). Linear regressions were calculated for (**a**) the rivers from the Po to Isonzo and (**b**) for all the NAd rivers.

The Timavo, Rižana and Dragonja rivers had low $Mg^{2+}$ concentrations that were not related to $HCO_3^-$, except for some very high values measured at the mouth of the Dragonja River that were influenced by saltwater intrusion. In contrast, the relationship between $Ca^{2+}$ and $HCO_3^-$ was significant for all the rivers with a ratio of 0.38 (Figure 3b, Table A1). Overall, these ratios indicated simultaneous weathering of dolomite and calcite in the river basins, from the Po to the Isonzo, and predominant calcite weathering in the smaller karst rivers in the easternmost part of the NAd.

### 4.3. Isotopic Signature and Weathering Intensity of the Riverine DIC

The stable isotopic composition of the DIC in freshwater ($\delta^{13}C_{DIC}$) is generally between $-25$ and $0‰$, depending on the main processes affecting the balance of the bicarbonates in the river basins. The weathering of carbonate rock produces $\delta^{13}C_{DIC}$ values around $0‰$, while the respiration of the soil and aquatic organic matter produces negative values (down to $-30‰$) due to the preferential release of isotopically light carbon ($^{12}C$) during the remineralization of organic matter. The primary production that preferentially assimilates $^{12}CO_2$, as well as the equilibration of the freshwater DIC with air $CO_2$ ($\delta^{13}C_{DIC} \approx -8‰$), generally results in progressively fewer negative $\delta^{13}C_{DIC}$ values in the rivers downstream. The $\delta^{13}C_{DIC}$ values in freshwater may also change seasonally due to the annual cycles of the biogeochemical processes in the river ecosystems [24,42].

The $\delta^{13}C_{DIC}$ data in the NAd rivers fell within a range of values from $-12.4$ to $-5.7‰$ (Figure 4, Table A1), and three different groups of data were identified.

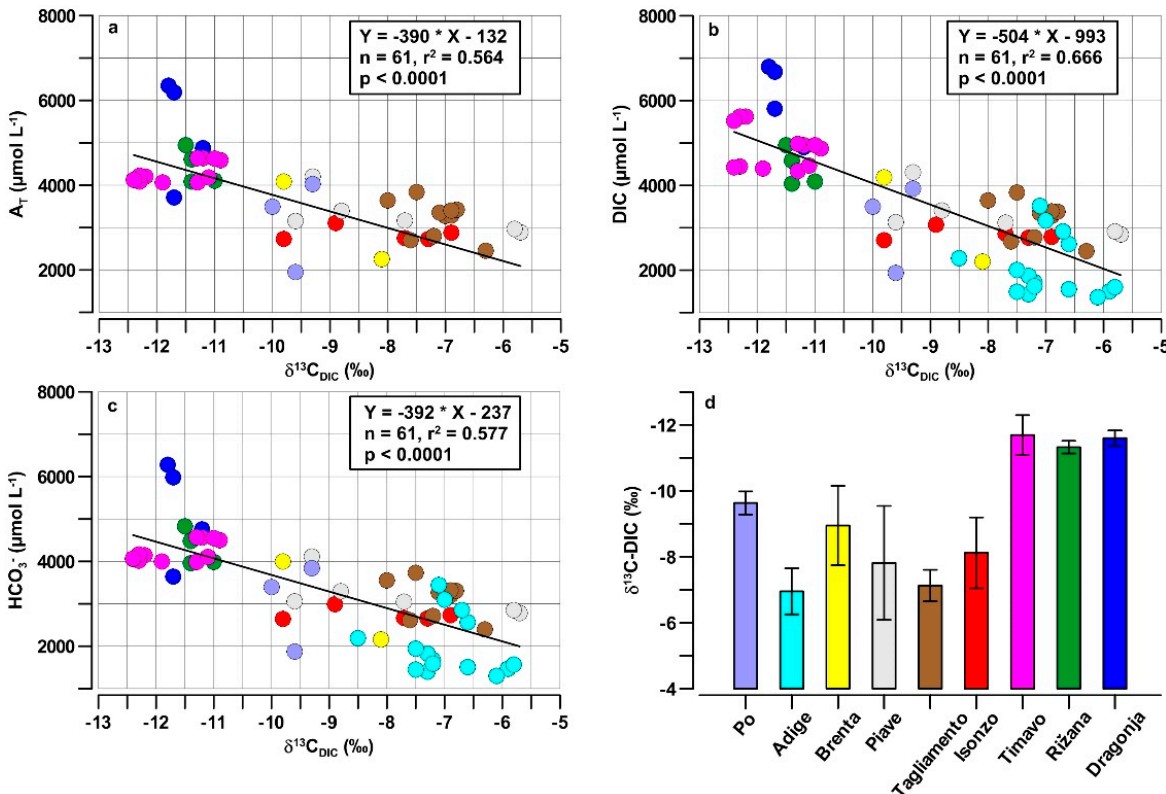

**Figure 4.** $\delta^{13}C_{DIC}$ values (‰) in river water versus the (**a**) $A_T$, (**b**) DIC and (**c**) $HCO_3^-$ concentrations (μmol L$^{-1}$). Panel (**d**) shows the mean and standard deviation of the $\delta^{13}C_{DIC}$ for each river.

1. The first group had the highest $A_T$, DIC and $HCO_3^-$ and the most negative $\delta^{13}C_{DIC}$ values, including the Dragonja, Rižana and Timavo. In the small rivers flowing in the karst catchments, the $\delta^{13}C_{DIC}$ values from the weathering of carbonate rocks were significantly shifted toward the negative values by the contribution of the remineralization of organic matter, which also produced the highest DIC and $HCO_3^-$ concentrations in freshwater.

2. The second group had a highly variable $A_T$, DIC and $HCO_3^-$ and medium to low values of the $\delta^{13}C_{DIC}$. This group included rivers such as the Brenta, Piave and Isonzo, which were probably influenced by a combination of processes along their river networks, such as the weathering of carbonate rocks, primary production and $CO_2$ exchange with the atmosphere, which increased the variability of these parameters. Minimal information was available on the $\delta^{13}C_{DIC}$ values in the Po River, but the data showed that these were intermediate values compared to the other NAd rivers, at least at the mouth of this large basin.

3. The third group presented a low $A_T$, DIC and $HCO_3^-$ and the least negative $\delta^{13}C_{DIC}$ values from the Adige and Tagliamento Rivers, which might have been due to the predominance of carbonate weathering in all the available sampling sites.

Despite the large variability in the river ecosystem characteristics of the NAd catchment, significant overall relationships were found between the $A_T$, DIC, $HCO_3^-$ and $\delta^{13}C_{DIC}$ (Figure 4a–c). These relationships were typical of sub-tropical temperate rivers, where the large DIC budgets in freshwater result from a variety of factors and not just from the decomposition of fresh organic carbon, as found in tropical rivers [43].

The NAd rivers basins were also characterized by a high runoff intensity compared to their extent, due to high annual precipitation and snowmelt in the surrounding mountain ranges [44]. Roy et al. [45] also found that interrelated factors such as the lithology, water residence time, mechanical erosion, etc. have a higher influence together than individually. Most studies on weathering in alpine regions concluded that enhanced mechanical erosion

in these environments also increases chemical weathering [46–48]. The stress on the mineral and an increased rock surface area create conditions under which minerals are more easily dissolved.

The global theoretical models of the $CO_2$ consumption in carbonate watersheds showed a value of the $A_T$ near 3000 μmol L$^{-1}$ determined from a best-fit line [49]. Although this value was reasonable as an average for all the NAd watersheds, many drainage basins have waters with much higher $HCO_3^-$ concentrations. The NAd river values ranged from 2.0 to 4.5 mmol $HCO_3^-$ per liter of runoff (Figure 5, Table A3).

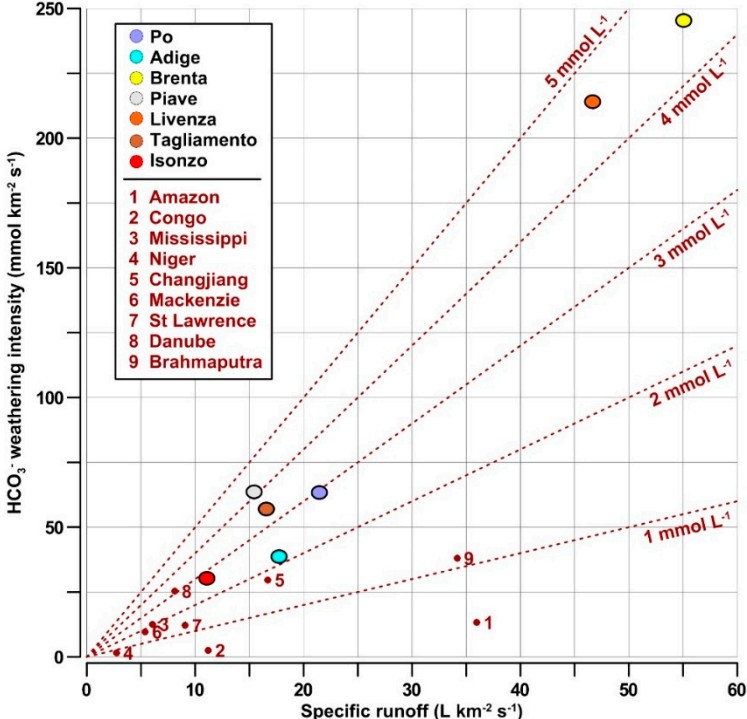

**Figure 5.** Weathering intensity of the $HCO_3^-$ (mmol km$^{-2}$ s$^{-1}$) as a function of the specific runoff (L km$^{-2}$ s$^{-1}$) in each drainage basin, in comparison to the most important rivers worldwide (data from [5]).

In particular, the Tagliamento, Piave, Brenta and Livenza rivers showed a high weathering intensity of the $HCO_3^-$ and DIC. This finding was consistent with the fewer negative $\delta^{13}C_{DIC}$ values found in some of these rivers, suggesting a stronger contribution of carbonate rock weathering to the total transport of the $HCO_3^-$. The Po River had a similar ratio for the $HCO_3^-$ weathering intensity to the specific runoff as the Danube (~3 mmol L$^{-1}$), although the Danube had a catchment area approx. ten times larger and a median annual flow approx. four times greater [30]. This result confirmed that the largest European rivers have high $HCO_3^-$ and DIC discharges compared to rivers in other continents (Figure 5, Table A3). The DIC weathering intensity is also high in Asian tropical rivers, which have the highest values (25.85 mmol C km$^{-2}$ s$^{-1}$) among tropical rivers worldwide (8.69 mmol C km$^{-2}$ s$^{-1}$) [50]. The DIC concentrations in Asian tropical rivers are lower than in the NAd rivers, possibly due to the abundant vegetation that reduces weathering and greater aquatic photosynthesis [50,51]. However, the contribution of European freshwater discharges should be better evaluated by including the data for medium and small catchments [43,52] and for submarine groundwaters [53,54].

The average isotopic signature of the NAd rivers was estimated to be −10.0 ± 1.7‰, while the Timavo, Rižana and Dragonja had −11.6 ± 0.4‰, using the equation in Section 3.3. These data are close to a $\delta^{13}C_{DIC}$ signature of −10‰, which is now considered to be representative of the DIC riverine inputs in ocean carbon cycle modeling [38]. Tropical rivers, such as the Amazon, Congo and Niger, often have strongly negative $\delta^{13}C_{DIC}$ values

due to the large contribution of fresh organic carbon decomposition to the freshwater $HCO_3^-$ fluxes. They are also characterized by a rather low $HCO_3^-$ weathering intensity, consistently with the global pattern of continental carbonate rocks and with their high discharges of organic carbon [5,24,43].

The variation in the $\delta^{13}C_{DIC}$ signature can be compared to the ratio $[H_2CO_3]/[HCO_3^-]$ in freshwater, as already proposed in the literature [49]. The plot of the $\delta^{13}C_{DIC}$ versus the ratio $[H_2CO_3]/[HCO_3^-]$ shows that two groups of rivers can be distinguished (Figure 6).

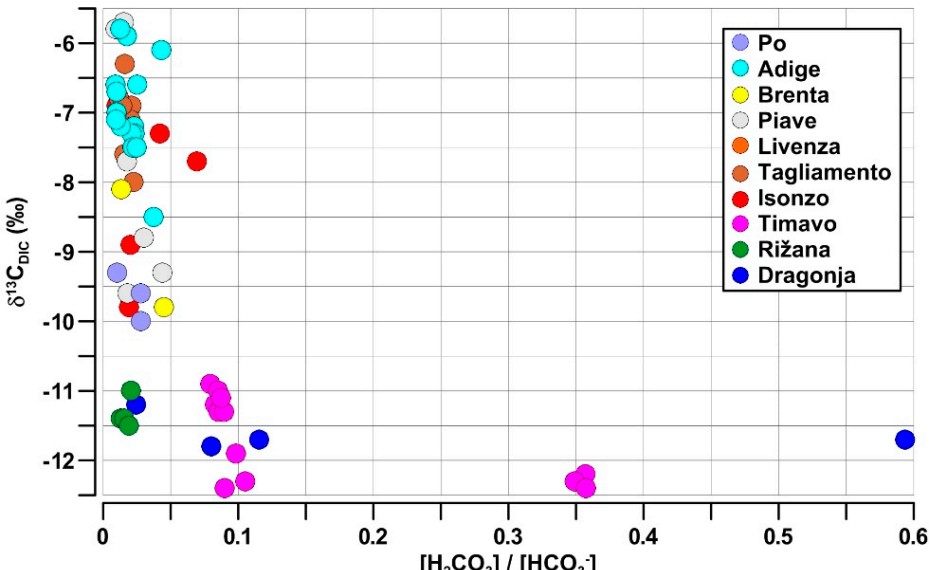

**Figure 6.** $\delta^{13}C_{DIC}$ values (‰) in the NAd rivers as a function of the $[H_2CO_3]/[HCO_3^-]$ ratio in freshwater.

1. Samples from the Timavo Rižana and Dragonja rivers, which show larger variations in the $[H_2CO_3]/[HCO_3^-]$ ratio, ranging between 0 and 0.6 and low values of the $\delta^{13}C_{DIC}$ (−11 to −12.5‰), indicating no exchange with atmospheric $CO_2$, but the presence of biogenic $CO_2$.
2. The second group, with a very low $[H_2CO_3]/[HCO_3^-]$ ratio (<0.1), exhibits greater variations of the $\delta^{13}C_{DIC}$ (−10 to −5‰), indicating different biogeochemical processes in the rivers such as an exchange with atmospheric $CO_2$, biogenic $CO_2$ and carbonate weathering.

### 4.4. Impact of the Riverine $A_T$ and DIC in the NAd

In the 2010s, the annual freshwater discharge of the Po River (27.2–75.1 km³ yr⁻¹) was approx. an order of magnitude higher than the other NAd rivers (0.6–10.7 km³ yr⁻¹). The year 2014 showed particularly high runoff, while 2017 was the driest year of this decade (Figure 7a, Table A2). For the other rivers, the freshwater discharge was in the order of the Adige > Brenta > Livenza > Piave. The transport of the $A_T$ and carbonate species followed the freshwater discharge, maintaining the same distinction between the Po and the other NAd rivers (Figure 7b–d, Table A2).

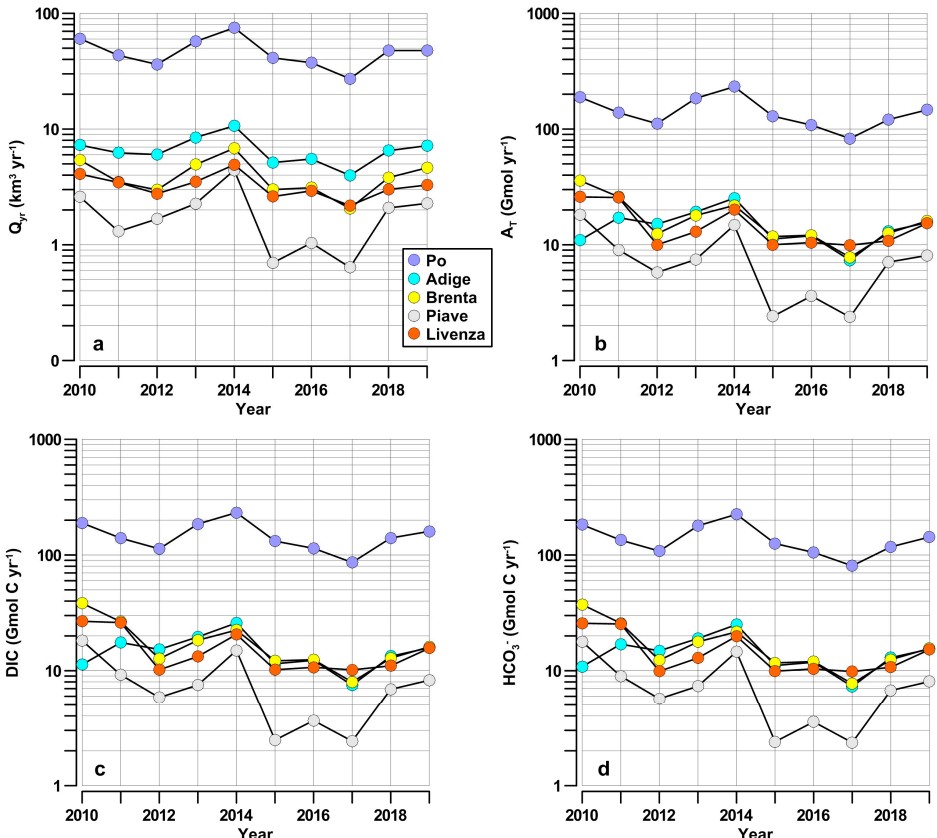

**Figure 7.** Annual discharges of (**a**) river water (Qyr; km$^3$ yr$^{-1}$), (**b**) total alkalinity (A$_T$; Gmol yr$^{-1}$), (**c**) dissolved inorganic carbon (DIC; Gmol C yr$^{-1}$) and (**d**) bicarbonate (HCO$_3$$^-$; Gmol C yr$^{-1}$) in the northern Adriatic by major rivers, in 2010–2019.

The annual transports of the A$_T$, DIC and HCO$_3$$^-$ by the Adige, Brenta and Livenza rivers were very similar, despite the differences in the freshwater discharges among these rivers. The Piave River almost always had the lowest freshwater and carbonate loads. The interannual variability of the freshwater and carbonate discharges was pronounced, reaching approx. 65% of these amounts when the extremely wet (2014) and dry (2017) years were compared.

On average, the transport of the A$_T$, DIC and HCO$_3$$^-$ through the Po River reached the values of 144 $\pm$ 43, 149 $\pm$ 41 and 140 $\pm$ 41 Gmol yr$^{-1}$, respectively in the 2010s (Figure 8, Table A2). The other NAd rivers were in the order of the Brenta, Livenza, Adige and Piave. Despite the limited availability of the data, the Tagliamento and Isonzo rivers were considered to be of minor importance at a basin scale, although they had a significant impact on the carbonate marine system at the sub-regional scale [24,55]. Historical data for the A$_T$ river concentrations in the 1970s were used by Copin Montegut [15] to estimate the A$_T$ discharge in the Mediterranean basin and reused by Cossarini et al. [18] for the Adriatic Sea. These estimates were made only for the alkalinity in a few rivers using old data. Our current estimate is 5–15% higher than that calculated on the basis of the Copin Montegut [15] data, primarily due to a higher A$_T$ concentration in the riverine waters. The NAd rivers contributed to 64% of the total A$_T$ discharged in the Adriatic basin, according to the estimate of Cossarini et al. [18] and to 12% of the total A$_T$ discharged by all Mediterranean rivers, according to the estimate of Copin Montegut [15], or to 23%, according to the estimate of Cossarini et al. [18].

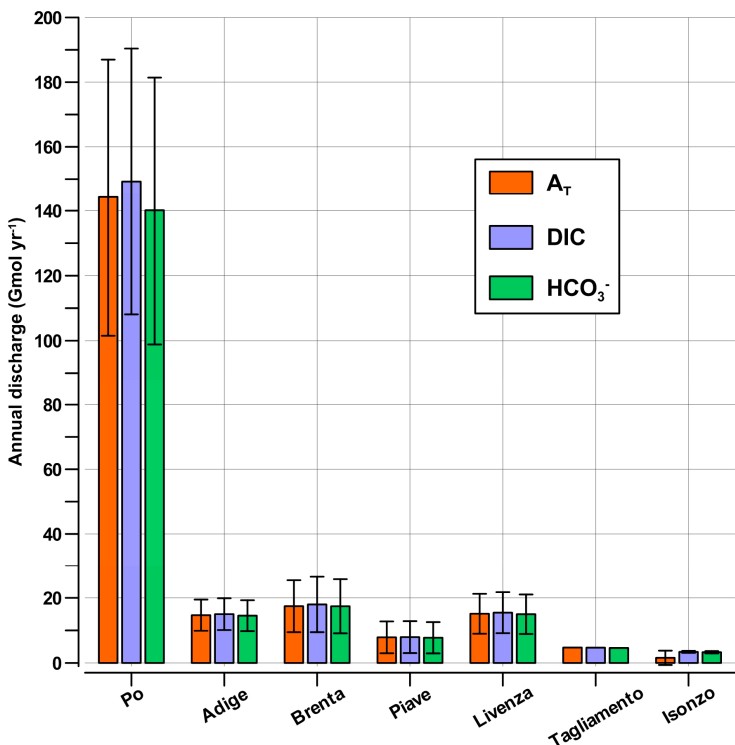

**Figure 8.** Mean annual transport of $A_T$, DIC and $HCO_3^-$ (Gmol yr$^{-1}$) from the NAd rivers. Data for the Tagliamento and Isonzo rivers were estimated for 2007 and 2011 only.

The mean DIC flux of the Po was slightly lower than the flux of the Rhône (162 G mol C yr$^{-1}$) [56], but almost four times lower than the flux of the Danube (629 G mol C yr$^{-1}$) [5]. The total DIC flux into the NAd represented 13–15% of the total DIC input by rivers in the Mediterranean Sea, estimated by Sempéré et al. [56].

The considered rivers did not represent the total discharge of carbonates in the NAd, which also consisted of small watercourses and freshwater discharges through coastal lagoons and submarine springs along the east coast. Nevertheless, a comparison can be made between the marine budgets of the $A_T$, DIC and $HCO_3^-$ in the NAd by calculating the turnover rates of these components. In the 2010s, the mean freshwater discharge of the NAd rivers was 65.7 km$^3$ yr$^{-1}$, which represented 24.7% yr$^{-1}$ of the seawater volume of the region shown in Figure 1 (i.e., 266 km$^3$). These rivers also discharged an average of 205.2 Gmol yr$^{-1}$ of $A_T$, 212.8 Gmol yr$^{-1}$ of DIC and 202.6 Gmol yr$^{-1}$ of $HCO_3^-$. Assuming the typical concentrations in seawater of $A_T$ = 2745.4 ± 34,5 µmol L$^{-1}$, DIC = 2428.4 ± 44.7 µmol L$^{-1}$ C and $HCO_3^-$ = 2181.9 ± 73.6 µmol L$^{-1}$ (see Section 3.3), the total budgets of these parameters in the NAd can be estimated to be 729.2, 645.0 and 579.5 Gmol, respectively. These data indicate that river discharges have an important influence on the carbonate system in the NAd, accounting for 28, 33 and 35% yr$^{-1}$ of the budget of these parameters in the sea, respectively (Table 2).

**Table 2.** Comparison between the volume of the North Adriatic (NAd) and the annual river discharge, as well as the budgets (Gmol) and mean annual river discharges (Gmol yr$^{-1}$) of the $A_T$, DIC and $HCO_3^-$. Turnover rates of all the parameters are expressed as percentage per year (% yr$^{-1}$).

|  | **Water** | **$A_T$** | **DIC** | **$HCO_3^-$** |
|---|---|---|---|---|
| Bulk NAd | 266 km$^3$ | 729.2 ± 9.2 Gmol | 645.0 ± 11.9 Gmol | 579.5 ± 19.5 Gmol |
| River discharge | 65.7 ± 24.5 km$^3$ yr$^{-1}$ 24.7 ± 9.2% yr$^{-1}$ | 205.2 ± 75.8 Gmol yr$^{-1}$ 28.1 ± 10.7% yr$^{-1}$ | 212.8 ± 72.6 Gmol yr$^{-1}$ 33.0 ± 11.9% yr$^{-1}$ | 202.6 ± 72.2 Gmol yr$^{-1}$ 35.0 ± 13.6% yr$^{-1}$ |

The Northern Adriatic region under consideration receives the largest runoff in the entire Mediterranean compared to its seawater volume, since it is a shallow continental shelf surrounded by an extremely large catchment area. As a result, the mean annual outflows of the total nitrogen and total phosphorus can be as much as seven times higher than the budgets of these biogenic elements in the NAd [28,29]. The present study showed, for the first time, the importance of continental inputs of the $A_T$, DIC and $HCO_3^-$ in the context of river discharge, as they account for approx. one-third of the marine budgets of these parameters at the annual scale. However, human activities can alter the terrestrial weathering and land surface hydrology, and thus, the $HCO_3^-$ fluxes in the rivers [14,57]. Pollution, acid rainfall and the use of high N fertilizers can increase the acidification of the soils and promote carbonate dissolution [5].

The potential effect of river discharges on ocean acidification in the NAd marine ecosystem can also be inferred from the composite parameter $[A_T - DIC]$, which is a property conservative with the mixing and almost linearly correlated to the changes in the pH and the saturation state of the calcium carbonate [58]. The river waters in the NAd often have higher concentrations of the $A_T$ and DIC than the seawater (Table 1), but their values of the $[A_T - DIC]$ ($-628$ to $46$ $\mu$mol L$^{-1}$) are often lower than the values usually found in seawater ($321.6 \pm 131.9$ $\mu$mol L$^{-1}$). Negative $[A_T - DIC]$ values characterized the Po, Brenta, Piave, Timavo and Dragonja, while the Adige, Livenza, Tagliamento Isonzo and Rižana had slightly positive values. Negative values are common to most of the rivers worldwide as they originate through the $CO_2$ sovrasaturation in freshwater systems [58]. This feature means that, in these cases, the river discharges can potentially decrease the pH in the NAd coastal waters [7,14,59]. However, it should also be kept in mind that large phytoplankton blooms occur in the river plumes due to the discharge of land-borne nutrients, often reaching eutrophic and hypertrophic conditions in the NAd [28,35]. Primary production can reduce the DIC concentrations, thereby increasing the pH in seawater.

River discharges are also characterized by a multiscale spatio-temporal variability due to the effects of climatic fluctuations and a continuous evolution of anthropogenic pressure in their catchments. For these reasons, more comprehensive monitoring of the carbonate system in freshwater ecosystems should be conducted to better model the future trends and variations of ocean acidification in the coastal areas [1,2].

## 5. Conclusions

This study shows that river discharge has an important influence on the carbonate balance and dynamics in the NAd, but it also suggests that more comprehensive monitoring of the carbonate system in the river basins should be conducted to better quantify the contributions of smaller watercourses, coastal lagoons, coastal groundwaters and submarine springs. Although the Po River was by far the largest contributor to the $A_T$ and DIC discharges, the impacts of the other sources of continental water cannot be ignored, at least at the sub-regional scale.

The NAd and its watershed is also region that is changing under the influence of climate trends and human activities. For these reasons, the river discharges of the $A_T$ and DIC might be easily altered in the future, having possible unexpected impacts on the seasonal and interannual variations of the carbonate system in the coastal waters.

The comparison between the inorganic carbon parameters, the stable isotopic composition of the DIC and the $Ca^{2+}$ and $Mg^{2+}$ concentrations showed that the carbonate system in the river drainage basins depends on a variety of factors, including rock and soil weathering, biological processes in the river networks, air and freshwater pollution and changes in the river regime. These factors, which are common in the temperate watersheds in industrialized regions, establish the NAd region as an important site for understanding the evolution of river–sea systems.

The following key points were highlighted in this study.

- Strong bicarbonate weathering occurred in these mountainous watersheds.

- Calcite weathering was prevalent in the Po, Isonzo, Timavo and Rižana catchments, while dolomite weathering was more important in the Adige, Piave and Livenza rivers.
- The rivers with an $A_T$ higher than the DIC concentration have the potential to buffer the NAd waters, while those, such as the Po River, have the potential to enhance the acidification process in the marine environment.
- The carbonate weathering intensity in the NAd watershed was among the highest and was approx. four to 35 times higher than the global average (7 meq km$^{-2}$ s$^{-1}$);
- Global climate models predicted that the precipitation pattern in southern Europe is likely to change. If the rivers in the NAd maintain a relatively unchanged concentration of carbonate mineral weathering products, the carbonate inputs to the coastal areas are expected to depend primarily on the overall changes in freshwater discharge.

**Supplementary Materials:** The following supporting information can be downloaded at: https://www.mdpi.com/article/10.3390/w15050894/s1, Key parameters measured in the NAd rivers are available in the file: Suppl-River-data-Tab S1.xlsx. Table S1: Physical and chemical parameters measured in the nortehrn Adriatic rivers.

**Author Contributions:** M.G. supervised the data mining and processing, M.G., N.O., S.T. processed the data set, M.G. and S.C. analyzed the data and prepared the figures, M.G., N.O., S.T. and S.C. wrote the paper, M.G. and S.C. performed the final editing of the paper. All authors have read and agreed to the published version of the manuscript.

**Funding:** This study was conducted under the RINGO Project "Readiness of ICOS for Necessities of Integrated Global Observations" (Grant agreement no. 730944) and the DANUBIUS Research Infrastructure (H2020—INFRA DEV 2). The research was partly funded by the Slovenian Research Agency under program P1-0143.

**Acknowledgments:** The authors thank the Agencies for the Environmental Protection (ARPA) of the Emilia Romagna, Veneto and Friuli Venezia Giulia regions for providing the physical and chemical data for the Italian rivers, AcegasAps for the Timavo River data and the Slovenian Environmental Agency (ARSO) for the Slovenian river data. The authors also thank the reviewers for their useful comments.

**Conflicts of Interest:** The authors declare that the research was conducted in the absence of any commercial or financial relationship that could be construed as a potential conflict of interest.

## Appendix A

**Table A1.** Physical and chemical parameters, including $\delta^{13}C_{DIC}$, measured in river waters (data source: Jožef Stefan Institute).

| River | Sample Site | Sampling Date dddd/mm/yy | Water Flow m$^3$ s$^{-1}$ | Specific Runoff L km$^2$ s$^{-1}$ | T °C | pH NBS Scale | EC µS cm$^{-1}$ | A$_T$ µmol L$^{-1}$ | Ca$^{2+}$ µmol L$^{-1}$ | Mg$^{2+}$ µmol L$^{-1}$ | Cl$^-$ µmol L$^{-1}$ | $\delta^{13}$C-DIC ‰ | DIC µmol L$^{-1}$ | HCO$_3^-$ µmol L$^{-1}$ | CO$_3$ $^{2-}$ µmol L$^{-1}$ | H$_2$CO$_3$ µmol L$^{-1}$ |
|---|---|---|---|---|---|---|---|---|---|---|---|---|---|---|---|---|
| Isonzo | Gradisca | 30/5/07 | 50.7 | 24.9 | 13.5 | 7.80 | 262 | 2736 | 1048 | 306 | 40 | −7.3 | 2769 | 2650 | 7 | 112 |
| Isonzo | Gradisca | 31/10/07 | 26.4 | 16.4 | 13.5 | 8.05 | 318 | 2715 | 1250 | 381 | 50 | - | 2699 | 2625 | 12 | 62 |
| Isonzo | Gradisca | 6/6/08 | 153.8 | 63.0 | 12.8 | 8.15 | 257 | 2732 | 1340 | 257 | 69 | −9.8 | 2704 | 2639 | 15 | 50 |
| Isonzo | Gradisca | 26/5/11 | 2.3 | 14.8 | 19.1 | 8.38 | 226 | 2877 | 1007 | 334 | 49 | −6.9 | 2788 | 2729 | 31 | 27 |
| Isonzo | Gradisca | 13/10/11 | 0.4 | 12.5 | 14.2 | 8.11 | 264 | 3103 | 1104 | 348 | 50 | −8.9 | 3070 | 2992 | 16 | 61 |
| Isonzo | Pieris | 30/5/07 | 50.7 | - | 15.0 | 7.57 | 274 | 2748 | 961 | 288 | 49 | −7.7 | 2852 | 2663 | 4 | 185 |
| Isonzo | Pieris | 31/10/07 | 26.4 | - | 11.3 | 8.03 | 336 | 3082 | 1365 | 411 | 63 | - | 3081 | 2990 | 12 | 77 |
| Dragonja | Mouth | 29/5/07 | 0.3 | 3.2 | 14.3 | 6.64 | 7490 | 3713 | 1688 | 5300 | 54,505 | −11.7 | 5809 | 3644 | 1 | 2164 |
| Dragonja | Mouth | 29/10/07 | 0.1 | 0.7 | 12.5 | 8.06 | 570 | 5119 | 2704 | 2387 | 22,085 | - | 5178 | 5035 | 24 | 117 |
| Dragonja | Mouth | 9/6/08 | 1.2 | 12.4 | 19.9 | 7.98 | 641 | 4871 | 2381 | 525 | 1395 | −11.2 | 4897 | 4757 | 23 | 116 |
| Dragonja | Mouth | 27/5/11 | 0.1 | 0.9 | 18.6 | 7.47 | 2225 | 6346 | 2846 | 2098 | 17,107 | −11.8 | 6794 | 6279 | 9 | 503 |
| Dragonja | Mouth | 13/10/11 | 0.0 | 0.1 | 17.6 | 7.33 | 12,560 | 6185 | - | - | - | −11.7 | 6674 | 5979 | 6 | 690 |
| Rižana | Bridge | 23/5/07 | 0.2 | 1.0 | 21.1 | 8.05 | 449 | 4108 | 1986 | 259 | 192 | −11.0 | 4087 | 3982 | 23 | 82 |
| Rižana | Bridge | 29/10/07 | 3.6 | 17.4 | 11.4 | 8.27 | 429 | 4397 | 2162 | 230 | 89 | - | 4377 | 4281 | 32 | 63 |
| Rižana | Bridge | 9/6/08 | 4.0 | 19.5 | 14.6 | 8.30 | 345 | 4081 | 2063 | 233 | 104 | −11.4 | 4040 | 3954 | 34 | 51 |
| Rižana | Bridge | 27/5/11 | 0.5 | 2.5 | 15.3 | 8.21 | 349 | 4612 | 1997 | 212 | 109 | −11.4 | 4585 | 4482 | 32 | 70 |
| Rižana | Bridge | 13/10/11 | 0.2 | 0.8 | 14.9 | 8.13 | 432 | 4948 | 2011 | 285 | 163 | −11.5 | 4946 | 4825 | 29 | 91 |
| Timavo | Spring 1 | 30/5/07 | 33.0 | 87.3 | 11.9 | 6.88 | 454 | 4200 | 2104 | 268 | 206 | −12.2 | 5625 | 4143 | 1 | 1479 |
| Timavo | Spring 1 | 31/10/07 | 10.0 | 26.5 | 13.6 | 7.36 | 464 | 4135 | 2155 | 282 | 195 | - | 4542 | 4072 | 4 | 465 |
| Timavo | Spring 1 | 10/6/08 | 23.0 | 60.8 | 12.1 | 7.41 | 373 | 4082 | 2065 | 267 | 190 | −12.3 | 4446 | 4018 | 4 | 422 |
| Timavo | Spring 1 | 26/5/11 | 15.0 | 39.7 | 12.5 | 7.51 | 332 | 4640 | 1843 | 352 | 208 | −11.2 | 4946 | 4561 | 6 | 378 |
| Timavo | Spring 1 | 13/10/11 | 12.0 | 31.7 | 13.6 | 7.52 | 397 | 4587 | 1749 | 380 | 236 | −10.9 | 4866 | 4502 | 6 | 356 |
| Timavo | Spring 2 | 30/5/07 | 33.0 | - | 11.9 | 6.89 | 455 | 4221 | 2080 | 262 | 203 | −12.3 | 5619 | 4163 | 1 | 1453 |
| Timavo | Spring 2 | 31/10/07 | 10.0 | - | 12.9 | 7.47 | 457 | 4121 | 2110 | 279 | 195 | −12.4 | 4427 | 4056 | 5 | 365 |
| Timavo | Spring 2 | 10/6/08 | 23.0 | - | 12.2 | 7.4 | 377 | 4034 | 2025 | 264 | 190 | - | 4400 | 3969 | 4 | 426 |
| Timavo | Spring 2 | 26/5/11 | 15.0 | - | 12.4 | 7.5 | 318 | 4631 | 1823 | 342 | 208 | −11.0 | 4945 | 4552 | 6 | 386 |
| Timavo | Spring 2 | 13/10/11 | 12.0 | - | 13.3 | 7.49 | 395 | 4065 | 1716 | 368 | 240 | −11.3 | 4333 | 3986 | 5 | 341 |
| Timavo | Spring 3 | 30/5/07 | 33.0 | - | 11.9 | 6.88 | 456 | 4124 | 2036 | 263 | 203 | −12.4 | 5520 | 4065 | 1 | 1453 |
| Timavo | Spring 3 | 31/10/07 | 10.0 | - | 13.0 | 7.39 | 467 | 4041 | 2124 | 280 | 198 | - | 4413 | 3979 | 4 | 429 |
| Timavo | Spring 3 | 10/6/08 | 23.0 | - | 15.5 | 7.41 | 392 | 4067 | 2019 | 263 | 190 | −11.9 | 4393 | 3995 | 5 | 392 |
| Timavo | Spring 3 | 26/5/11 | 15.0 | - | 12.0 | 7.48 | 321 | 4642 | 1839 | 346 | 212 | −11.3 | 4980 | 4564 | 6 | 409 |
| Timavo | Spring 3 | 13/10/11 | 12.0 | - | 13.4 | 7.48 | 393 | 4183 | 1752 | 372 | 241 | −11.1 | 4468 | 4104 | 5 | 358 |

**Table A1.** *Cont.*

| River | Sample Site | Sampling Date dddd/mm/yy | Water Flow m³ s⁻¹ | Specific Runoff L km² s⁻¹ | T °C | pH NBS Scale | EC µS cm⁻¹ | $A_T$ µmol L⁻¹ | Ca²⁺ µmol L⁻¹ | Mg²⁺ µmol L⁻¹ | Cl⁻ µmol L⁻¹ | δ¹³C-DIC ‰ | DIC µmol L⁻¹ | HCO₃⁻ µmol L⁻¹ | CO₃²⁻ µmol L⁻¹ | H₂CO₃ µmol L⁻¹ |
|---|---|---|---|---|---|---|---|---|---|---|---|---|---|---|---|---|
| Tagliamento | Invilino | 3/6/08 | 35.0 | - | 15.8 | 8.19 | 598 | 2452 | 2993 | 886 | 52 | −6.3 | 2449 | 2393 | 17 | 39 |
| Tagliamento | Tolmezzo | 18/10/11 | 38.2 | - | 12.6 | 8.15 | 653 | 3275 | 2737 | 926 | 88 | −7.0 | 3286 | 3205 | 19 | 60 |
| Tagliamento | Pioverno | 2/6/11 | 23.0 | 16.0 | 14.8 | 8.09 | 429 | 3289 | 1930 | 729 | 98 | −6.9 | 3283 | 3198 | 17 | 67 |
| Tagliamento | Pioverno | 18/10/11 | 38.2 | 11.7 | 10.5 | 8.14 | 492 | 3352 | 2007 | 829 | 95 | −7.1 | 3351 | 3267 | 17 | 66 |
| Tagliamento | Dignano | 3/6/08 | 38.4 | - | 14.8 | 8.22 | 413 | 2704 | 1842 | 695 | 203 | −7.6 | 2675 | 2615 | 19 | 40 |
| Tagliamento | Dignano | 2/6/11 | 23.0 | - | 17.5 | 8.32 | 477 | 3426 | 2097 | 762 | 92 | −6.8 | 3375 | 3304 | 32 | 39 |
| Tagliamento | Dignano | 18/10/11 | 38.2 | - | 11.3 | 8.29 | 527 | 3408 | 2183 | 845 | 94 | −6.9 | 3384 | 3311 | 26 | 47 |
| Tagliamento | Madrisio | 6/6/08 | 36.0 | - | 16.0 | 8.20 | 428 | 2803 | 1785 | 710 | 94 | −7.2 | 2771 | 2709 | 19 | 43 |
| Tagliamento | Madrisio | 30/5/11 | 26.1 | - | 16.2 | 8.05 | 433 | 3836 | 1917 | 831 | 90 | −7.5 | 3836 | 3733 | 19 | 83 |
| Tagliamento | Madrisio | 20/10/11 | 91.0 | - | 13.6 | 8.07 | 494 | 3643 | 1817 | 812 | 74 | −8.0 | 3644 | 3546 | 18 | 79 |
| Piave | Provagna | 17/6/08 | 100.0 | - | 14.0 | 7.94 | 337 | 3392 | 1529 | 654 | 80 | −8.8 | 3411 | 3299 | 12 | 99 |
| Piave | Provagna | 1/6/11 | 26.7 | - | 15.0 | 8.23 | 290 | 2893 | 1308 | 562 | 44 | −5.7 | 2846 | 2783 | 20 | 42 |
| Piave | Provagna | 18/10/11 | 7.3 | - | 10.6 | 8.49 | 429 | 2969 | 1817 | 753 | 57 | −5.8 | 2910 | 2849 | 34 | 26 |
| Piave | San Dona | 6/6/08 | 168.0 | 42.8 | 15.4 | 8.15 | 336 | 3158 | 1531 | 542 | 88 | −9.6 | 3127 | 3053 | 19 | 55 |
| Piave | San Dona | 30/5/11 | 68.0 | 13.0 | 16.5 | 8.15 | 334 | 3162 | 1461 | 582 | 77 | −7.7 | 3125 | 3052 | 19 | 54 |
| Piave | San Dona | 20/10/11 | 13.8 | 9.5 | 13.2 | 7.78 | 457 | 4201 | 1724 | 735 | 107 | −9.3 | 4303 | 4111 | 10 | 180 |
| Brenta | Valstagna | 1/6/11 | 100.5 | - | 12.8 | 8.40 | 144 | 1742 | 686 | 230 | 48 | - | 1685 | 1651 | 16 | 18 |
| Brenta | Valstagna | 20/10/11 | 67.8 | - | 11.5 | 8.32 | 256 | 2255 | 985 | 373 | 66 | −8.1 | 2203 | 2157 | 17 | 29 |
| Brenta | Limena | 30/5/11 | 104.9 | 39.9 | 16.2 | 8.18 | 215 | 2766 | 963 | 316 | 86 | - | 2713 | 2651 | 17 | 44 |
| Brenta | Limena | 20/10/11 | 67.8 | 22.0 | 13.1 | 7.77 | 424 | 4086 | 1567 | 592 | 214 | −9.8 | 4183 | 3993 | 10 | 180 |
| Po | Pollesella | 8/6/08 | 3160.0 | 45.1 | 18.8 | 7.94 | 286 | 1949 | 1054 | 307 | 310 | −9.6 | 1932 | 1872 | 8 | 52 |
| Po | Pollesella | 30/5/11 | 1030.0 | 14.7 | 23.7 | 8.33 | 477 | 4024 | 1652 | 562 | 621 | −9.3 | 3921 | 3837 | 44 | 40 |
| Po | Pollesella | 20/10/11 | 873.0 | 12.5 | 14.5 | 7.97 | 469 | 3490 | 1518 | 540 | 557 | −10.0 | 3500 | 3392 | 14 | 94 |
| Adige | San Michele | 17/6/08 | 404.2 | - | 10.8 | 8.05 | 164 | 1566 | 751 | 287 | 86 | −6.6 | 1547 | 1502 | 6 | 38 |
| Adige | San Michele | 31/5/11 | 259.3 | - | 13.7 | 8.18 | 158 | 1533 | 662 | 257 | 82 | −5.9 | 1496 | 1462 | 9 | 26 |
| Adige | San Michele | 19/10/11 | 175.2 | - | 8.1 | 8.12 | 234 | 1739 | 867 | 347 | 116 | −7.2 | 1716 | 1670 | 8 | 38 |
| Adige | Mattarello | 15/6/08 | 484.7 | - | 10.8 | 7.82 | 151 | 1351 | 703 | 257 | 79 | −6.1 | 1358 | 1299 | 3 | 56 |
| Adige | Chizzola | 15/6/08 | 484.7 | - | 11.2 | 8.08 | 164 | 1455 | 756 | 275 | 98 | −7.3 | 1433 | 1394 | 6 | 33 |
| Adige | Chizzola | 1/6/11 | 260.8 | - | 13.4 | 8.11 | 166 | 1903 | 741 | 273 | 107 | −7.3 | 1870 | 1822 | 9 | 38 |
| Adige | Chizzola | 19/10/11 | 175.2 | - | 9.6 | 7.89 | 269 | 2266 | 1010 | 396 | 150 | −8.5 | 2280 | 2192 | 6 | 82 |
| Adige | Peri | 15/6/08 | 484.7 | - | 11.6 | 8.11 | 169 | 1515 | 780 | 285 | 93 | −7.5 | 1490 | 1451 | 7 | 31 |
| Adige | Peri | 30/5/11 | 312.6 | - | 13.9 | 8.14 | 174 | 1817 | 720 | 269 | 103 | - | 1780 | 1737 | 10 | 33 |
| Adige | Peri | 19/10/11 | 175.2 | - | 8.7 | 8.08 | 262 | 2018 | 941 | 399 | 138 | −7.5 | 2000 | 1943 | 8 | 48 |
| Adige | Avisio | 17/6/08 | 404.2 | - | 12.7 | 8.32 | 163 | 1666 | 743 | 228 | 147 | −7.2 | 1619 | 1585 | 13 | 21 |
| Adige | Avisio | 31/5/11 | 259.3 | - | 16.3 | 8.3 | 169 | 1647 | 732 | 229 | 117 | −5.8 | 1596 | 1563 | 13 | 20 |
| Adige | Avisio | 19/10/11 | 175.2 | - | 9.7 | 8.5 | 329 | 2687 | 1314 | 449 | 151 | −6.6 | 2619 | 2565 | 30 | 23 |
| Adige | Ala | 15/6/08 | 484.7 | - | 12.3 | 8.44 | 240 | 2993 | 1049 | 679 | 29 | −6.7 | 2914 | 2854 | 31 | 28 |
| Adige | Ala | 1/6/11 | 260.8 | - | 14.2 | 8.43 | 216 | 3257 | 929 | 593 | 27 | −7.0 | 3165 | 3100 | 35 | 30 |
| Adige | Ala | 19/10/11 | 175.2 | - | 9.5 | 8.48 | 293 | 3601 | 1055 | 696 | 28 | −7.1 | 3511 | 3440 | 38 | 33 |

**Table A2.** Annual discharges of freshwater, total alkalinity ($A_T$), dissolved inorganic carbon (DIC) and bicarbonates ($HCO_3^-$) by the northern Adriatic rivers. Averages (AV) and standard deviations (SD) are reported.

| Year | Freshwater km³ yr⁻¹ | $A_T$ Gmol yr⁻¹ | DIC Gmol C yr⁻¹ | $HCO_3^-$ Gmol C yr⁻¹ | Year | Freshwater km³ yr⁻¹ | $A_T$ Gmol yr⁻¹ | DIC Gmol C yr⁻¹ | $HCO_3^-$ Gmol C yr⁻¹ |
|---|---|---|---|---|---|---|---|---|---|
| Po River | | | | | Piave River | | | | |
| 2010 | 60.47 | 188.52 | 188.99 | 182.90 | 2010 | 2.60 | 18.16 | 18.20 | 17.80 |
| 2011 | 43.26 | 138.46 | 139.74 | 134.43 | 2011 | 1.30 | 9.04 | 9.21 | 8.96 |
| 2012 | 36.21 | 111.32 | 113.07 | 108.11 | 2012 | 1.67 | 5.77 | 5.85 | 5.71 |
| 2013 | 57.16 | 184.26 | 185.38 | 178.91 | 2013 | 2.27 | 7.46 | 7.52 | 7.35 |
| 2014 | 75.09 | 232.17 | 232.63 | 224.85 | 2014 | 4.40 | 14.77 | 14.95 | 14.61 |
| 2015 | 41.20 | 129.24 | 131.83 | 125.60 | 2015 | 0.70 | 2.41 | 2.47 | 2.39 |
| 2016 | 37.44 | 108.25 | 114.35 | 105.27 | 2016 | 1.04 | 3.61 | 3.65 | 3.55 |
| 2017 | 27.23 | 82.79 | 86.19 | 80.65 | 2017 | 0.64 | 2.37 | 2.42 | 2.35 |
| 2018 | 47.68 | 120.92 | 140.20 | 117.63 | 2018 | 2.08 | 3.09 | 7.18 | 7.01 |
| 2019 | 47.70 | 146.61 | 160.06 | 142.96 | 2019 | 2.28 | 8.10 | 8.27 | 8.05 |
| AV | 47.34 | 144.25 | 149.25 | 140.13 | AV | 1.90 | 7.48 | 7.97 | 7.78 |
| SD | 13.11 | 42.73 | 41.15 | 41.29 | SD | 1.06 | 5.09 | 4.90 | 4.80 |
| Adige River | | | | | Livenza River | | | | |
| 2010 | 7.27 | 10.93 | 11.30 | 10.85 | 2010 | 4.08 | 25.92 | 26.75 | 25.70 |
| 2011 | 6.23 | 17.14 | 17.53 | 16.96 | 2011 | 3.46 | 25.61 | 26.07 | 25.37 |
| 2012 | 6.03 | 15.08 | 15.27 | 14.88 | 2012 | 2.76 | 10.00 | 10.18 | 9.90 |
| 2013 | 8.47 | 19.20 | 19.61 | 19.00 | 2013 | 3.50 | 12.98 | 13.27 | 12.88 |
| 2014 | 10.67 | 25.29 | 25.90 | 25.06 | 2014 | 4.92 | 20.11 | 20.60 | 19.86 |
| 2015 | 5.14 | 11.23 | 11.50 | 11.09 | 2015 | 2.61 | 10.02 | 10.19 | 9.92 |
| 2016 | 5.51 | 11.96 | 12.31 | 11.85 | 2016 | 2.92 | 10.46 | 10.70 | 10.37 |
| 2017 | 3.98 | 7.35 | 7.50 | 7.29 | 2017 | 2.18 | 9.92 | 10.17 | 9.84 |
| 2018 | 6.51 | 13.10 | 13.34 | 12.97 | 2018 | 3.01 | 10.84 | 11.03 | 10.71 |
| 2019 | 7.19 | 15.55 | 15.79 | 15.36 | 2019 | 3.29 | 15.34 | 15.67 | 15.22 |
| AV | 6.70 | 14.68 | 15.00 | 14.53 | AV | 3.27 | 15.12 | 15.46 | 14.98 |
| SD | 1.77 | 4.79 | 4.89 | 4.74 | SD | 0.74 | 6.13 | 6.30 | 6.07 |

**Table A2.** *Cont.*

| Year | Freshwater km$^3$ yr$^{-1}$ | $A_T$ Gmol yr$^{-1}$ | DIC Gmol C yr$^{-1}$ | HCO$_3{}^-$ Gmol C yr$^{-1}$ | Year | Freshwater km$^3$ yr$^{-1}$ | $A_T$ Gmol yr$^{-1}$ | DIC Gmol C yr$^{-1}$ | HCO$_3{}^-$ Gmol C yr$^{-1}$ |
|---|---|---|---|---|---|---|---|---|---|
| Brenta River | | | | | | | | | |
| 2010 | 5.39 | 35.91 | 38.28 | 37.27 | | | | | |
| 2011 | 3.49 | 25.98 | 26.52 | 25.74 | | | | | |
| 2012 | 2.98 | 12.45 | 12.71 | 12.33 | | | | | |
| 2013 | 4.93 | 17.92 | 18.35 | 17.79 | | | | | |
| 2014 | 6.82 | 21.87 | 22.46 | 21.68 | | | | | |
| 2015 | 3.00 | 11.77 | 12.18 | 11.68 | | | | | |
| 2016 | 3.11 | 12.09 | 12.45 | 12.01 | | | | | |
| 2017 | 2.06 | 7.79 | 7.98 | 7.71 | | | | | |
| 2018 | 3.81 | 12.60 | 12.86 | 12.48 | | | | | |
| AV | 3.95 | 17.60 | 18.20 | 17.63 | | | | | |
| SD | 1.40 | 8.40 | 8.96 | 8.74 | | | | | |

**Table A3.** Comparison of different river fluxes and weathering intensities of dissolved inorganic carbon and bicarbonates in this study (2010–2019) and in the literature.

| River Name | Basin Area km$^2$ | Water Flow km$^3$ yr$^{-1}$ | Specific Runoff L km$^{-2}$ s$^{-1}$ | Carbonate in Soil % | Basin Latitude Degree | HCO$_3{}^-$ µmol L$^{-1}$ | DIC Flux Gmol yr$^{-1}$ | HCO$_3{}^-$ Flux Gmol yr$^{-1}$ | HCO$_3{}^-$ Weathering Intensity mmol km$^{-2}$ s$^{-1}$ | DIC Weathering Intensity mmol km$^{-2}$ s$^{-1}$ | Reference |
|---|---|---|---|---|---|---|---|---|---|---|---|
| Po | 70,091 | 47.30 | 21.4 | - | 45 N | 2929 | 148.0 | 139.82 | 63.25 | 66.98 | This study |
| Adige | 11,954 | 6.70 | 17.8 | - | 45 N | 2131 | 15.0 | 14.53 | 38.55 | 39.80 | This study |
| Brenta | 2280 | 3.95 | 55.0 | - | 45 N | 3325 | 18.2 | 17.63 | 245.23 | 253.11 | This study |
| Piave | 3899 | 1.90 | 15.4 | - | 46 N | 3629 | 8.0 | 7.78 | 63.27 | 64.84 | This study |
| Livenza | 2222 | 3.27 | 46.7 | - | 46 N | 4309 | 15.5 | 14.98 | 213.73 | 220.66 | This study |
| Tagliamento | 2582 | 1.35 | 16.5 | - | 46 N | 3231 | 4.7 | 4.62 | 56.70 | 57.89 | This study |
| Isonzo | 3452 | 1.20 | 11.0 | - | 46 N | 2723 | 3.4 | 3.30 | 30.29 | 31.30 | This study |
| Amazon | 5,854,000 | 6642 | 36.0 | 3.9 | 2 S | 369 | 2450 | 2450.90 | 13.28 | 13.27 | [5] |
| Congo | 3,699,000 | 1308 | 11.2 | 10.1 | 4 S | 224 | 293 | 292.99 | 2.51 | 2.51 | [5] |
| Mississippi | 3,203,000 | 610 | 6.0 | 18.1 | 36 N | 2074 | 1265 | 1265.14 | 12.52 | 12.52 | [5] |
| Niger | 2,240,000 | 193 | 2.7 | 6.3 | 10 N | 550 | 106 | 106.15 | 1.50 | 1.50 | [5] |
| Changjiang | 1,794,000 | 944 | 16.7 | 44.0 | 30 N | 1780 | 1680 | 1680.32 | 29.70 | 29.69 | [5] |
| Mackenzie | 1,713,000 | 290 | 5.4 | 20.6 | 64 N | 1800 | 522 | 522.00 | 9.66 | 9.66 | [5] |
| St Lawrence | 1,267,000 | 363 | 9.1 | 24.9 | 47 N | 1339 | 486 | 486.06 | 12.16 | 12.16 | [5] |
| Danube | 788,000 | 202 | 8.1 | 14.5 | 48 N | 3115 | 629 | 629.23 | 25.32 | 25.31 | [5] |
| Brahmaputra | 583,000 | 628 | 34.2 | 33.8 | 25 N | 1114 | 700 | 699.59 | 38.05 | 38.07 | [5] |
| Asian tropical rivers | 11,342,854 | 8694 | 24.3 | - | 30 N-30 S | 1064 * | 9241 | - | - | 25.85 | [57] |

* Concentration of DIC.

**Table A4.** Calcium and magnesium concentrations (mmol L$^{-1}$) in the riverine waters. Average (AV), standard deviation (SD) and number of measurements (N) are reported.

| River | Period | N | Ca$^{2+}$ AV | Ca$^{2+}$ SD | Mg$^{2+}$ AV | Mg$^{2+}$ SD |
|---|---|---|---|---|---|---|
| | | | mmol L$^{-1}$ | mmol L$^{-1}$ | mmol L$^{-1}$ | mmol L$^{-1}$ |
| Po | 2010–2019 | 82 | 1.45 | 0.23 | 0.48 | 0.09 |
| Adige | 2010–2019 | 28 | 1.00 | 0.34 | 0.45 | 0.33 |
| Brenta | 2010–2018 | 15 | 1.05 | 0.32 | 0.38 | 0.13 |
| Piave | 2010–2019 | 15 | 1.62 | 0.17 | 0.82 | 0.46 |
| Livenza | 2010–2019 | 7 | 1.52 | 0.27 | 0.68 | 0.11 |
| Tagliamento | 2008–2019 | 47 | 1.93 | 0.30 | 0.85 | 0.10 |
| Isonzo | 2007; 2008; 2011; 2016 | 18 | 1.20 | 0.26 | 0.36 | 0.14 |
| Timavo | 1998–2003;2006–2012; 2016 | 173 | 1.96 | 0.15 | 0.31 | 0.05 |
| Rižana | 2007, 2008, 2011 | 5 | 2.04 | 0.06 | 0.24 | 0.03 |
| Dragonja | 2007, 2008, 2011 | 3 | 2.40 | 0.45 | 2.58 | 1.72 |

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
