# Peer review of "Elevated River Inputs of the Total Alkalinity and Dissolved Inorganic Carbon in the Northern Adriatic Sea"

_water, doi:10.3390/w15050894_

Round 1

Reviewer 1 Report

Dear authors

Please specify what conductivity and pH meters were used (at least name and country), what calibration solutions were used and what pH scale you use.

L149 - "combination electrodes". Do you mean combined glass electrodes with thermistor?  

In general, there are no complaints about the work. Please check the text again in the context of the English language

Author Response

Please, find enclosed the replay to reviewer comments in PDF document.

Best regards,

The correponding authors

Reviewer 2 Report

I am an oceanographer trained in chemistry. I find it difficult to tolerate measurements without QA/QC. This manuscript does not even give precision or accuracy for the parameters measured. This raises doubt about the quality of the data although new data are always welcome.

The manuscript is straightforward and can be improved easily by describing the QA/QC and attending to the following minor comments:

1. It is confusing seeing the statement in the abstract that land-derived DIC can counteract ocean acidification by absorbing CO2 from the atmosphere. In seawater, any addition of DIC tends to reduce the ability of the seawater to absorb CO2. The better indicator is DIC/TA, and reducing it increases the ability to counteract ocean acidification.

2. The studied rivers are all very small so it may not be proper to compare the results with large rivers. For instance, the HCO3 vs runoff should be compared with small rivers such as those published in other marginal seas(e.g., Huang et al., JGR Biogeosciences, 2017, 122, doi 10.1002/2016JG003701; Current Opinion in Env. Sus., 2012,4,162).

Author Response

Please, find enclosed the replay to the comments of the Reviewer 2 in PDF document.

Best regards,

The corresponding authors

Reviewer 3 Report

I find it a well written and illustrated study. It corresponds to a case study of notable interest: a shallow marine sector with abundant fluvial discharge from catchment basins with abundance of carbonates and significant human activity. It is not possible to speak here of novel methodologies, but of a case study of the FTAA that can serve as a model for others in more or less similar areas or with less abundance of data (in this regard, the additional information provided is also noteworthy). The discussion and conclusions seem relevant and appropriate to me, as well as the bibliographical references.

Author Response

Please, find enclosed the replay to the comments of the Reviewer 3 in PDF document.

Best regards,

The corresponding authors

Reviewer 4 Report

First, I believe this is one of the interesting environment subject to study. I would to congratulate the authors for the quality of this work. Indeed, Before definitively adopting this article, I would like that the authors take into account the following remarks and suggestions:

1. why this "The Po River being the main contributor" is considered? and can you mention some of the anthropogenic factors that you sited in the paper!? 

2. Can you discuss more if there are any pollutions sources in the study area!? 

3. Some results of rivers are not cited in figure or tables

a) Table A1 in Appendix section :nothing about  Livenza river, why? 

b) Figure 2: nothing about Dragonja, why?

c) Figure 4: nothing about  Dragonja, why?

d) Figure 5: nothing about Timavo, Rižana and Dragonja, why?

e) Figure 7: nothing about Isonzo, Timavo, Rižana and Dragonja, why?

f) Figure 8:  nothing about Timavo, Rižana and Dragonja, why?

4. There is a problem regarding the references, for example in the Line 182; you find reference 52 after 28... please check the hall document and update the references numbering. 

Author Response

Please, find enclosed the replay to the comments of the Reviewer 4 in PDF document.

Best regards,

The corresponding authors

Round 2

Reviewer 2 Report

I am glad that the revised manuscript replied to my comments properly.